# Genomic Diversity of SARS-CoV-2 in Algeria and North African Countries: What We Know So Far and What We Expect?

**DOI:** 10.3390/microorganisms10020467

**Published:** 2022-02-18

**Authors:** Taha Menasria, Margarita Aguilera

**Affiliations:** 1Department of Applied Biology, Faculty of Exact Sciences and Natural and Life Sciences, University of Larbi Tebessi, Tebessa 12002, Algeria; 2Department of Microbiology, Faculty of Pharmacy, University of Granada, 18071 Granada, Spain

**Keywords:** North Africa, SARS-CoV-2, Algeria, genomics, clade, Omicron, mutation annotation

## Abstract

Here, we report a first comprehensive genomic analysis of SARS-CoV-2 variants circulating in North African countries, including Algeria, Egypt, Libya, Morocco, Sudan and Tunisia, with respect to genomic clades and mutational patterns. As of December 2021, a total of 1669 high-coverage whole-genome sequences submitted to EpiCoV GISAID database were analyzed to infer clades and mutation annotation compared with the wild-type variant Wuhan-Hu-1. Phylogenetic analysis of SARS-CoV-2 genomes revealed the existence of eleven GISAID clades with GR (variant of the spike protein S-D614G and nucleocapsid protein N-G204R), GH (variant of the ORF3a coding protein ORF3a-Q57H) and GK (variant S-T478K) being the most common with 25.9%, 19.9%, and 19.6%, respectively, followed by their parent clade G (variant S-D614G) (10.3%). Lower prevalence was noted for GRY (variant S-N501Y) (5.1%), S (variant ORF8-L84S) (3.1%) and GV (variant of the ORF3a coding protein NS3-G251V) (2.0%). Interestingly, 1.5% of total genomes were assigned as GRA (Omicron), the newly emerged clade. Across the North African countries, 108 SARS-CoV-2 lineages using the Pangolin assignment were identified, whereby most genomes fell within six major lineages and variants of concern (VOC) including B.1, the Delta variants (AY.X, B.1.617.2), C.36, B.1.1.7 and B.1.1. The effect of mutations in SAR-CoV-2 genomes highlighted similar profiles with D614G spike (S) and ORF1b-P314L variants as the most changes found in 95.3% and 87.9% of total sequences, respectively. In addition, mutations affecting other viral proteins appeared frequently including; N:RG203KR, N:G212V, NSP3:T428I, ORF3a:Q57H, S:N501Y, M:I82T and E:V5F. These findings highlight the importance of genomic surveillance for understanding the SARS-CoV-2 genetic diversity and its spread patterns, leading to a better guiding of public health intervention measures. The know-how analysis of the present work could be implemented worldwide in order to overcome this health crisis through harmonized approaches.

## 1. Introduction

Officially, in late December 2019, the World Health Organization (WHO) was notified by the Chinese Health Authorities of pneumonia cases of unknown etiology detected in Wuhan City, Hubei Province [1], which could mark the emergence of a novel and serious threat to public health. On 7 January 2020, researchers from the Shanghai Public Health Clinical left and School of Public Health reported the isolation of a new type of coronavirus (novel coronavirus, nCoV) [2] and a preliminary analysis of the Wuhan virus sequence (WH-Human_1.fasta.gz), [Genbank/NCBI release (MN908947.1)] suggesting a possible zoonotic origin [3]. 

Between 10 and 15 January 2020, findings of unexplained pneumonia in a Shenzhen family cluster confirmed the presence of the novel coronavirus, and suggested possible sustained human-to-human transmission [4], despite the fact that the extent of this mode of transmission is unclear. Since the first report, other territories, areas and countries outside China have reported cases among travelers returning from different countries and the epidemiologic picture is changing on a daily basis. As of 30 January 2020, the WHO officially declared that the new coronavirus outbreak [coronavirus disease (COVID-19)] is a public health emergency of international concern and later on March 11 declared it as a pandemic, acknowledging what has seemed clear—that the virus will likely spread to all countries on the globe, possibly intensified by high population density, socio-demographic profiles, host immune responses, and viral genetics [5].

The virus causes a respiratory illness that is often severe, and at least 5.3 million deaths have been reported globally [6]. The current estimates of the case fatality rate of SARS-CoV-2 at any time point of analysis should be interpreted with caution since the outcome of the emerging COVID-19 is as yet unknown. 

The coronaviruses are incredibly diverse, encompassing a broad spectrum of animal and human enveloped RNA viruses. Prior to 2003, CoV commonly caused mild but occasionally severe community-acquired acute respiratory infections in humans [7]. The recent outbreak takes us back to other coronavirus endemics caused by SARS-CoV (Severe Acute Respiratory Syndrome-related coronavirus) in China in 2002 and the zoonosis caused by the Middle East Respiratory Syndrome coronavirus (MERS-CoV), which first emerged in Saudi Arabia in 2012 before spreading to other countries [8]. In fact, the epidemiology reports of MERS-CoV and SARS-CoV infections have been characterized by recurrent zoonotic leaks from the known primary animal reservoir (bats) to intermediary sources ‘Dromedary camels or civet cats’ and human to human transmission with fatality rates close to 9.5% and 35%, respectively [9,10]. The disease outbreak highlights the hidden reservoir of the viruses in wild animals and their potential to spill out into humans. 

As of 25 February 2020, a case of COVID-19 was reported in the first Member State of the AFRO Region [11] and the questions were raised of whether more human cases will occur in Africa and what measures would African countries take to mitigate the SARS-CoV-2 threat? Eighteen months later, data as reported by national authorities on August 2021, have shown large community spread of COVID-19 cases with more than seven and a half million confirmed cases [12] and at least 198,000 deaths attributed to the virus.

Generally, new changes and variants of SARS-CoV-2 constantly emerge and circulate around the world during the COVID-19 pandemic [13]. Most changes have little to no impact on the virus’ properties. However, some changes may affect the virus’ properties such as transmission, the associated disease severity, the performance of vaccines, therapeutic medicines, diagnostic tools, or other public health and social measures [14]. More recently, a Variant Classification scheme that defines three classes of SARS-CoV-2 variants has been developed in order to prioritize global monitoring and research, and ultimately to inform the ongoing response to the COVID-19 pandemic including: (i) variants of concern (VOC); (ii) variants of interest (VOI); and (iii) variants of severe consequence (VOHC) [15]. The variants of concern (VOC) are B.1.1.7 (WHO labeled Alpha) first identified in the UK, B.1.351 (WHO labeled Beta) in South Africa, P.1 (Gamma) in Brazil and B.1.617.2, AY.X (Delta) in India. VOIs include variants B.1.427/9 (Epsilon), B.1.525 (Eta), B.1.526 (Iota), B.1.617.1 (Kappa), C.37 (Lambda) and Mu (B.1.621). On November 24, 2021, a new variant, B.1.1.529, was first reported to the WHO by South Africa, identified from an immunocompromised patient in Johannesburg and, based on the evidence presented indicative of a detrimental change in COVID-19 epidemiology, the WHO designated B.1.1.529 a VOC, named Omicron [15]. Currently, there are no SARS-CoV-2 variants that rise to the level of high consequence (VOHCs).

As COVID-19 vaccines become available and are implemented, genomic surveillance, together with real-time tracing and data-sharing networks, has become a valuable tool to improve understanding of SARS-CoV-2 transmission patterns and epidemic dynamics. Analysis of these data played a key role in the response to the pandemic by tracking the global spread of novel SARS-CoV-2 variants, leading to a greater understanding of COVID-19 outbreaks around the world. By using a comprehensive genomic analysis, the current study aims to provide information, for the first time, about the geographic distribution of SARS-CoV-2 genomic lineages and potential diversification pathways of the virus in Algeria and North African countries. The circulation of these variants has broad epidemiological implications for public health, including ongoing vaccination efforts.

## 2. Materials and Methods

### 2.1. Epidemiological Dynamics and Genomic Data Processing

The complete genome sequences of SARS-CoV-2 isolates from Algeria and North African countries, including Egypt, Libya, Morocco and Tunisia, were retrieved from the EpiCoV database of the GISAID initiative [16]. As of 20 December 2021, 2599 genomes were downloaded and only viruses affecting human hosts were selected, removing low-quality sequences (>5% NNNs) and using only full-length sequences (>29,000 nt). In total, 1669 complete, high coverage genome sequences from the dataset were selected to investigate the genetic characterization (Table 1). 

Daily updates on the number of confirmed new cases of COVID-19 in Algeria were analyzed up of February 2020 (for 20 months) from publicly released data provided by the Algerian Ministry of Health, Population and Hospital Reform (https://www.sante.gov.dz/) (accessed on 20 December 2021).

### 2.2. Sequence Alignment and Phylogenetic Analysis of Algerian Genomes

For the local Algerian virus comparison, thirty-six sequences were first aligned using a multiple sequence alignment algorithm (MAFFT v7. 471) [17]. The maximum likelihood tree was reconstructed with the IQ-TREE server using the general time-reversible (GTR) model with rate heterogeneity (GTR + G) and 1000 ultrafast bootstrap repetitions (http://www.iqtree.org accessed on 20 December 2021) [18]. The SARS-CoV-2 genomes were classified into lineages using the PANGOLIN web application (Phylogenetic Assignment of Named Global Outbreak LINeages) (https://pangolin.cog-uk.io accessed on 20 December 2021) [19]. The viral clades were assigned by the Nextclade tool (https://clades.nextstrain.org/ accessed on 20 December 2021) [20] and through the UShER web interface from the University of California, Santa Cruz (https://genome.ucsc.edu/cgi-bin/hgPhyloPlace accessed on 20 December 2021). Viral clades were defined on the basis of available genomes sharing the same pattern of mutations. The Algerian population was comparatively evaluated against the Wuhan reference genome (NC_045512.2-Wuhan-Hu-1) obtained from NCBI GenBank. Quality checks of the sequences and evaluation of genetic distance were performed in MEGA software version 6 [21] and the final dataset was displayed using Interactive Tree of Life (iTOL) v.4 (https://itol.embl.de/ accessed on 20 December 2021) [22].

### 2.3. Mutation Signature and Clade Assignment Analysis

The Nextclade web tool (https://clades.nextstrain.org accessed on 20 December 2021) and the online COVID-19 genome annotator ‘coronapp’ [23] were used to perform mutation signature calling and SNP profile defining of total genome sequences from North African countries by checking amino acid substitutions, deletion or insertion mutations on specific regions, including; spike surface glycoprotein (S), polyprotein 1ab (nsp1-nsp16), structural proteins (S, E, M, and N) and other accessory proteins. In addition, genomic lineages and clades were inferred by GISAID and PANGOLIN databases according to the nomenclature system at the time of data collection. 

## 3. Results

### 3.1. Epidemiology of SARS-CoV-2 in Algeria and North Africa

By 20 December 2021, over 2,500,000 SARS-CoV-2 cumulative cases had been reported in North African countries (Figure 1A) of which 214,592 cases were confirmed in Algeria with more than 6190 deaths attributed to the virus and a case fatality ratio (CFR) of 2.88%. In addition, Algeria’s western neighbor, Morocco, registered the highest rate of positivity among the North African countries with 34.3% of total cases (952,814) followed by Tunisia, Libya, Egypt, and Sudan with 721,031; 381,749; 375,330 and 45,112 of positive cases, respectively (Table 1).

The first confirmed positive case of SARS-CoV-2 infection in Algeria was reported on 25 February 2020, initially among international travelers until flights were stopped in March 2020. Immediately after the first case, the country experienced several waves of the pandemic. The second wave of viral introductions occurred between October and December 2020 and included migrants returning from Europe, followed by a third wave of rapid growth in Mid July and August 2021 in terms of the daily incidence of positivity and deaths (Figure 1B). 

### 3.2. Phylogenetic Analysis of SARS-CoV-2 Genomes in Algeria

Of the 85 available sequences, 36 genomes that met the quality criteria for analysis (>90% coverage) were used to construct a maximum-likelihood phylogenetic tree. As presented in Figure 2, the phylogenetic analysis is in support of the PANGOLIN lineages assignment of which the analyzed SARS-CoV-2 genomes belonged to six different B lineages (B.1, B.1.1, B.1.159, B.1.36, B.1617.2 and BA.1) and three other A lineages (AY.20, AY.44 and AY122) clustered together compared to the reference NC_045512.2-Wuhan-Hu-1.

Algeria has little diversity in variant mapping, which is not surprising given limitations to whole genome sequencing. The Nextclade analysis revealed that seventeen of the 36 SARS-CoV-2 genomes belonged to the Delta clade (21J), with the rest being part of clades 20A, 20B and 20C. More recently, two genomes were submitted in December 2021 to the EpiCoV database and were grouped with 21K. Furthermore, GISAID analysis showed that the selected sequences belong to four high-level phylogenetic groups including G, GH, GR and GK with 16 genomes (44.4%) as part of the GK (Delta) clade and nine others (25.8%) of the GH (Beta) clade. The time course of the phylogenetic analysis and clade distribution showed that clades G (Variant S-D614G), GH (Variant ORF3a-Q57H) and GR (Variant N-G204R) were the most prevalent in the first and second waves of viral introductions. However, this was no longer the case in early May and mid-July 2021, in which clade GK took over. Expanded phylogenetic analysis was conducted to examine the genetic divergence of Algerian samples against global representative SARS-CoV-2 genomes present in the Nextstrain database. The mutation-resolved ML phylogenetic tree confirmed the PANGO and Nextclade lineages assigned, since 17 genomes grouped with the 21J representatives, six (16.6%) with the 20A clade, four (11.1%) belong to the 20B clade, eight (22.2%) with the 20C sequences and two (5.4%) with the 21K (Omicron) clade (Figure 2). 

### 3.3. Distribution of SARS-CoV-2 Lineages and Clades in North Africa

The variants from 1669 retrieved genomes were clustered in 108 SARS-CoV-2 lineages using the Pangolin web services tool, whereby most samples fell within six dominant lineages (Figure 3 and Figure 4) including B.1, the Delta variants (AY.X, B.1.617.2), C.36, B.1.1.7 and B.1.1 with 20.2%, 19.1%, 17.8%, 6.9%, and 5.9%, respectively. The Nextstrain clade assignment revealed that the analyzed genomes formed fifteen distinct clades with 20A, 20D and 21J (Delta) being the most common with 26.4%, 24.1% and 18.7% respectively, followed by 20I (Alpha) and 20B with 8.1%–6.7% each. Analysis of the distribution of SARS-CoV-2 clades in North African countries showed that clade GR was the most frequently identified with 25.9% among the total genomes, followed by GH and GK (19.9%–19.6%) and their parent clade G (10.3%). Other less common clades including S and L were also identified in 3.1% and 1.7% of the analyzed sequences, respectively. Furthermore, about 5% of the genomes were not clustered into any of the major clades. Interestingly, 1.5% of total genomes were assigned as GRA (Omicron), the newly emerged clade.

#### 3.3.1. Egypt

A total of fifty-two lineages have been identified by the Pangolin phylogenetic classification, of which five were most prevalent including C.36 (30.6%), followed by B.1 (25.2%), C.36.3 (7.2%), B.1.1 (5.1%) and B.1.617.2 with 5.1% of total analyzed genomes (Figure 4). These lineages are associated with clades GR, GH, and GK respectively. The Nextclade analysis revealed that 41.2% of the 971 SARS-CoV-2 genomes belonged to clade 20D, followed by 20A (29.1%) and 21J (Delta) (13.1%) with the rest being part of clades 19B, 20B, 20C, 21I (Delta), 20I (Alpha, V1) and only one genome classified as 21K (Omicron) submitted in December 12, 2021, isolated from a hospitalized male patient aged 67 years.

#### 3.3.2. Libya

Eight Pango lineages were identified in the analyzed sequences from Libya. The B.1.525 was identified as the dominant sublineage with 57.9% of total analyzed SARS-CoV-2 genomes retrieved from GISAID. The lineage (A) ranked in second place with 18.4% followed by the sublineages B.1 and B.1.1.7 with 7.9% and 5.3% respectively (Figure 3). 

#### 3.3.3. Morocco

Forty-six Pango lineages and sublineages were observed in Morocco, of which the B.1.17 presented the most prevalent lineage of 104/352 sequenced genomes (29.5%) followed by B.1 (73 genomes, 20.7%), B.1.1 (9.4%), B.1.528 (6.0%) and the new lineage (BA.1-Omicron) with (6.3%) (22/352) of analyzed genomes. Morocco have reported the most diverse clades (ten clades) among north African countries, of which clade G was the highest with 25.0%, followed by GRY (22.2%), GR (19.0%), GH (11.1%), GK (10.5 %), GRA (6.3%), GV (3.4%), O (1.7%), L (0.6%) and V with 0.3% of total analyzed genomes.

#### 3.3.4. Sudan

Nine Pango lineages were identified from Sudan, of which the A.9 and B.1.351 present the most prevalent lineages with 25.8% each followed by B.1 with 12.9% of total analyzed genomes. In addition, four different clades were reported with GH (51.6%) and S (35.5%) as dominant compared to GR, the clade with 9.7%. 

#### 3.3.5. Tunisia

The variants from 241 analyzed genomes clustered in twenty-six SARS-CoV-2 lineages and eleven clades (Figure 3), whereby most samples fell within four dominant lineages of the total for AY.122 (Delta) with 57.7%, B.1.160 (9.5%), B.1.177 (8.7%) and B.1.1 (3.3%). Among the studied genomes, clades GK, GH, GV, and GR were found to be dominant with 58.5%, 19.9%, 9.1% and 7.5% respectively (Figure 3).

### 3.4. Phylogenetic Analysis of Omicron Genome Sequences from North Africa

A total of 25 genome sequences were obtained from GISAID, collected between October and December 2021. Spatiotemporal phylogenetic analyses were conducted using the complete genomes available at the time, with two genomes from Algeria, one genome from Egypt and 22 genomes from Morocco. As is shown Figure 5A, the global phylogeny of SARS-CoV-2 sequences (Delta and Omicron) from North Africa (as of 20 December 2021) showed that Omicron sequences (21K) could have been a progeny of the nextclade 20B. The global subtrees (Figure 5B) showed evidence of different geographic origins of Omicron lineage. The majority of analyzed sequences were closely related with BA.1 sequences from England and the remained genomes were related to sequences from Scotland and United States suggesting multiple introductions of SARS-CoV-2 variants into North African countries. In addition, the phylogenetic analysis showed that the BA.1 genomes formed monophyletic clusters indicating local transmission of Omicron lineage in Morocco compared to the two Algerian sequences.

### 3.5. Genomic Variation and Mutation Signature

The retrieved SARS-CoV-2 genomes from North African countries were compared with the reference NC_045512.2-Wuhan-Hu-1 and, as expected, significant numbers of non-synonymous and synonymous mutations were detected. The annotated mutations, event by event, are summarized in Figure 6. The analysis of 1669 SARS-CoV-2 genomes have highlighted a total of 42,685 mutation events compared to the reference (Appendix A). A high prevalence of single-nucleotide polymorphisms (SNPs) was noted with 26,532 (62.17%) events over indels (insertion or deletions) with 2.31% and 0.014%, respectively. Furthermore, 11,222 events of silent SNPs falling in coding regions were detected, representing 26.30% of the total events. Overall, the C>T transition presents the most common events accounting for 42.63% with 18,198 in total, followed by G>T transversion at 16.81% with 7178 occurrence, A>G transition with 4449 events (10.42%) and G>A transition with 2071 (6.83%) of all observed viral mutations. 

The effects of mutations on the protein sequences of SARS-CoV-2 highlighted similar profiles in the North African countries with a mutation affecting the nucleotide adenosine in position (23,403) transformed into a guanosine (A23403G) causing a D614G spike (S) variant as the most common amino acid change occurring in 1553 (95.27%) of total SARS-CoV-2 genomes. A similar occurrence was also detected for the nucleotide cytosine in position (3037) (C3037T) affecting 1433 (87.91%) genomes causing an amino acid changing mutation in P314L, affecting the NSP12 (non-structural protein 12) and the viral RNA-dependent RNA polymerase (Figure 6 and Figure 7). Two other silent mutations were noted including C241T (92.57%) and C14408T (87.91%), targeting the 5′UTR and the NSP3 (a viral predicted phosphoesterase) in position 14408. In addition, mutations affecting other protein sequences appeared frequently including; N:RG203KR and N:G212V (Nucleocapsid protein N) with (636 genomes, 39.01%) and (339 genomes, 20.79%), respectively, M:I82T (Membrane protein) (411 genomes, 25, 25%), NSP3:T428I (phosphoesterase, papain-like proteinase) (400, genomes, 24.53%), ORF3a protein (ORF3a:Q57H) (350 genomes, 21, 47%), S:N501Y (Spike protein) (163 genomes, 10.0%), and E:V5F (Envelope protein) with a lower occurrence (2.21%) of total SARS-CoV-2 genomes (Figure 7).

### 3.6. Variant Analysis of Omicron SARS-CoV-2 Genomes

The analysis of the genetic polymorphism of Omicron SARS-CoV-2 genomes compared to the Wuhan-Hu-1 reference genome revealed variable mutations between viruses. A total of 1455 mutation events were noted with a high prevalence of single-nucleotide polymorphisms (SNPs) (914, 62.8%) and events followed by silent SNPs (235, 16.15%) over deletion with 5.85% of total events (Appendix A). The frequent mutation events observed for Omicron genomes are summarized in Table 2.

The amino acid substitutions (D614G, D614G, D796Y, T547K, N856K, N679K, N969K, P681H, L981F) in the spike protein, P314L, A1892T, T492I, I189V and A1892T in the non-structural proteins (NSP3, NSP4, NPS6, and NSP12b) occurred in 100% of analyzed SARS-CoV-2 genomes. In addition, fifteen other substitutions affect the spike protein including Q954H, N764K, H655Y, K417N, G339D, N211K, Q493R, S371L, S373P, S375F, S477N, T478K, E484A, N440K and G446S were detected as the most frequent mutation events in more than 84% of total genomes. Other mutations were found in 24/25 analyzed sequences distributed in the nonstructural proteins (nsp3, nsp4, nsp6, nsp14) which had the highest number of variants in the analyzed samples, followed by Nucleocapsid proteins (N), Membrane (M) protein (Table 2) and Open Reading Frame proteins (ORF3a, ORF7a, ORF7b, ORF8, ORF9b) (Appendix A).

## 4. Discussion

Despite substantial advances, the implementation of genomic surveillance remains a challenge for most African countries where access to whole genome sequencing is limited. Since the first description of the SARS-CoV-2 sequence in late 2019 [3], an exponentially increasing number of virus genomes have been reported across the globe with over ten million complete genomes deposited in the GISAID (https://www.gisaid.org accessed on 20 December 2021) and Genbank (https://www.ncbi.nlm.nih.gov/sars-cov-2/ accessed on 20 December 2021) databases (accessed on 20 December 2021). Nonetheless, SARS-CoV-2 genome sequence data from Africa constitute less than 0.7% with 70,421 sequences in genome repositories. 

Due to the naturally expanding genetic diversity of SARS-CoV-2 viruses, extensive molecular surveillance and efforts to understand the patterns of the global spread of the pandemic have been introduced including the three main nomenclatures, PANGO lineages (PANGO, Phylogenetic Assignment of Named Outbreak LINeages) [19], Nextstrain clades [20] and GISAID classification. While PANGO lineages provide more detailed outbreak cluster information, the other two nomenclatures offer broad geographical and temporal clade trends.

This paper presents the first insight into a comprehensive analysis of genome sequences of SARS-CoV-2 circulating in Algeria and North African countries. The data of 1669 SARS-CoV-2 genomes submitted to the EpiCoV GISAID database as of 20 December 2021 were analyzed with respect to genomic clades and their geographic distribution. The results revealed the presence of different clades and variants as defined by GISAID, PINGOLIN and Nextclade tools that could be involved in the varied exacerbation of symptoms and disease severity in local residents. 

As of 25 February 2020, a case of COVID-19 was reported in the first Member State of the AFRO Region leading the Algerian and neighboring health authorities to set up a response plan with rapid implementation to prevent and control SARS-CoV-2 spreading [24,25]. Despite the restrictions and the lockdown measures applied in most North African countries, the virus continued to spread from one region to another [26], and evolved with numerous genetic variants being associated with higher infectivity [27]. So far, the retrieved SARS-CoV-2 genomes were clustered into twelve major clades, as defined by the GISAID database, and at least 108 pingolineages, with six dominant variants including B.1, the Delta variants (AY.X, B.1.617.2), C.36, B.1.1.7 and B.1.1. Clades GR, GH and GK were the most frequently identified among the analyzed genomes, followed by G, GRY, GV and O clades, with lower prevalence confirming the heterogeneity of circulating strains. Interestingly, 1.5% of total genomes were assigned as GRA (Omicron), the newly emerged clade.

Early on in the first outbreak, the SARS-CoV-2 genomes were classified in two major lineages, named the European superclade A (also referred to L) and the East Asian superclade B (referred to S) [28,29], and later several sublineages in the GISAID nomenclatures have been introduced including V, G, GH, GR, GV and GRY clades based on marker mutation and phylogenetic analysis [30] (https://www.gisaid.org/ accessed on 20 December 2021). 

Globally, the G clade and its derivatives GH, GR, and GV are the most common clades amongst the sequenced SARS-CoV-2 genomes [31]. Mercatelli and Giorgi [23] reported that GISAID clades G, GR and GV are prevalently present in Europe with relatively higher COVID-19 cases, deaths and CFRs, while the clades GH and GR have been mostly observed in the Americas, the top ranked continents with respect to CFR and local disease epidemiology parameters. 

The dynamics of SARS-CoV-2 spreading in North Africa was not so different from that which was observed worldwide, with first and second waves dominated by viruses belonging to clades 20A and 20B, followed by a third wave linked to the circulation of variants characterized by an increase in the number of severe forms of COVID-19, leading to more deaths. Similarly, a study that investigated SARS-CoV-2 sequences collected in the Eastern Mediterranean Region found that more than 65.8% of the viruses belong to clades 20A, 20B, and 20C (GISAID clades GR, GH, G and GV) [32]. Similarly, genome sequencing of SAR-CoV2 isolated from Egyptian patients showed that most of the sequences can be assigned clades G/GR/GH/O (as per GISAID system) [33]. In addition, genomic surveillance applied to SARS-CoV-2 transmission in Morocco [34] between March and May 2020, revealed different aspects of the epidemic with the introduction of SARS-CoV-2 strains from different European countries where most genomes fell within Clades 20A, 20B with different mutation patterns giving rise to the diversity of SARS-CoV-2 lineages reported in this study. 

New changes and variants of SARS-CoV-2 constantly emerge as long as ongoing transmission persists causing major epidemics in the United Kingdom (UK) [19], Brazil [35,36], and South Africa [37]. The Delta variant (PANGO lineage: B.1.617.2), first detected in India, has spread quickly across the world, and is designated a variant of concern by the World Health Organization likely due to higher transmissibility prior to wild type infection, estimated to be about 60% more transmissible than the Alpha variant [38]. 

Recently, the Omicron variant (B.1.1.529) has been primarily of concern after the Delta variant due to its large number of mutations (26 to 32) in the genome compared with other variants, especially in the spike protein, many of which are located within the receptor binding domain (RBD) [39], known or predicted to contribute to escape from neutralizing antibodies and existing countermeasures. Recently, Omicron (B.1.1.529) was predicted to be associated with a rapid increase of COVID-19 cases (https://www.who.int/news/item/28-11-2021-update-on-omicron) (accessed on 30 December 2021). In a short period, the circulation of Omicron has been found in at least 65 countries and territories with thousands of confirmed cases (https://www.gisaid.org/hcov19-variants/ accessed on 20 December 2021).

The D614G spike mutation characterizes the G clade and its derivate has spread exponentially across the world and become rapidly the most prevalent lineage worldwide [40], occurring in over 92% of total analyzed genomes in this study. However, the A23403G mutation leading to the D614G spike (S) variant was found to be located in a heavily glycosylated residue in the viral spike, was implicated in increased infectiveness and allows fast spreading of the virus during the COVID-19 pandemic compared to the wild type variant Wuhan-Hu-1 [41].

It is worth mentioning that the Spike D614G mutation accompanies other frequent mutation sites in the ORF1ab (NSP3:C3037T, NSP3:T428I and NSP12:C14408T) region, the mutation at position 241 (C241T) targeting the 5′UTR, as well as the mutations at positions 203 and 212 in the Nucleocapsid protein (N:RG203KR, N:G212V), in the receptor binding domain (RBD) of Spike (S:N501Y), and in the ORF3a protein (ORF3a:Q57H). Generally, Spike D614G and ORF1b-P314L variants are consistently related and co-occur in all geographic locations with increasing frequency [42]. The spike glycoprotein region mediates the infection of target cells through binding to its cognate receptor angiotensin converting enzyme 2 (ACE2) and initiating viral–host fusion and replication [43]. This region is reported to be the most essential for viral attachment and entry into the host cells [44,45]. Therefore, ACE2 expression in different tissues and interactions with SARS-CoV-2 are critical for the infection’s progression to severe coronavirus disease 2019 (COVID-19) [46]. The P314L mutation in NSP12 (RNA-dependent polymerase) may play a causal role in viral replication, therefore enhancing its transmission ability and infectivity [44]. Moreover, extragenic SNPs in 5′ UTR:C241T may also affect the folding of the ssRNA and influence the replication rates of SARS-CoV-2 as it is found to occur most prominently [47]. Comparative genomic analysis of SARS-CoV-2 genomes revealed multiple crucial mutations to the Spike gene including K417N, K417T, E484K, N501Y, A570D, D614G, P681H, T716I, S982A and D1118H, which may aggravate the severity of SARS-CoV-2 more than the wild type variant, and potentially raise the concern of vaccine efficacy against novel strains [41,48]. 

The broad SARS-CoV-2 lineage diversity circulating in North African countries could intensify the impact of the pandemic in the region, affecting the efficacy of vaccines and displaying reduced antibodies neutralization, even reducing the reliability of diagnosis schemes including the current primary method of detecting SARS-CoV-2 (Reverse transcription-quantitative polymerase chain reaction (RT-qPCR)) [49,50]. However, and within a very short period of time, research applied to COVID-19 diagnosis has advanced with ever-increasing knowledge and inventions, in adapting available virus detection technologies and exploiting the power of interdisciplinary research to design novel diagnostic tools to improve detection efficiency [51,52]. Given the epidemiological behaviors, current evidence supports that VOCs, including Delta and the newly emerged Omicron variant, have rapidly escalated, becoming predominant in the globe and replacing previously circulating variants (https://nextstrain.org/ncov/gisaid/global) (accessed on 12 February 2022), adding up to a complex epidemiological scenario. 

Compared with other variants and the early identified SARS-CoV-2 strains, the high frequency of mutations in the spike sequence of the Omicron variant raises concern about potential immune escape and its impact remains to be determined [53]. However, a complete experimental evaluation of Omicron might take weeks or even months. Large-scale case-control studies are essential for investigating clinical severity and the current situation must lead national governments to place a higher priority on timely collection and analysis. In fact, COVID-19 severity varies enormously depending on the country, the prevalence of vaccination, the population’s characteristics and medical management guidelines [54]. 

## 5. Conclusions

Despite the presence of some limitations in the study, such as the absence of clinical data on patients, as well as unbalanced sample sizes among the analyzed countries, the data provide valuable information about the SARS-CoV-2 clades circulating in North African countries and may help inform the dynamics of the disease for better control measures and appropriate public health action as the pandemic spreads in Africa. Analysis of SARS-CoV-2 sequences highlighted, for the first time, the changing pattern of circulating SARS-CoV-2 lineages in Algeria and North Africa between February 2020 and December 2021. Distinct lineages of SARS-CoV-2 contributing to three separate waves of infections reflective of the epidemiological pattern were identified, leading to the detection of previously major circulating variants of concern (VOC) in addition to the newly emerged Omicron variant. 

As is known, the African region is characterized by the largest infectious disease burden and the weakest public health infrastructures, which can be explained by the fact that a large population is vulnerable due to conflict, poor socio-economic status, food insecurity and limited access to better health services. Furthermore, the prolonged humanitarian crises facilitate the spread of the actual disease within and between countries as well as causing extensive deterioration of health. Given the current epidemic and limited understanding of the epidemiology of this disease, the coronavirus poses a serious challenge for the continent and the emergence of a serious health threat highlights the need to support African countries with ‘Weaker Health Systems’. 

## Figures and Tables

**Figure 1 microorganisms-10-00467-f001:**
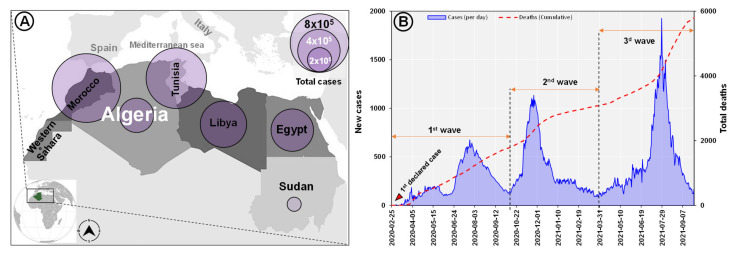
Total COVID-19 cases distribution in North African countries (**A**) and Algeria (daily new cases and cumulative deaths) (https://africacdc.org/covid-19/ accessed on 20 December 2021), (**B**) Graph based on a data source available online at (https://covid19.sante.gov.dz/carte/) (accessed on 20 December 2021).

**Figure 2 microorganisms-10-00467-f002:**
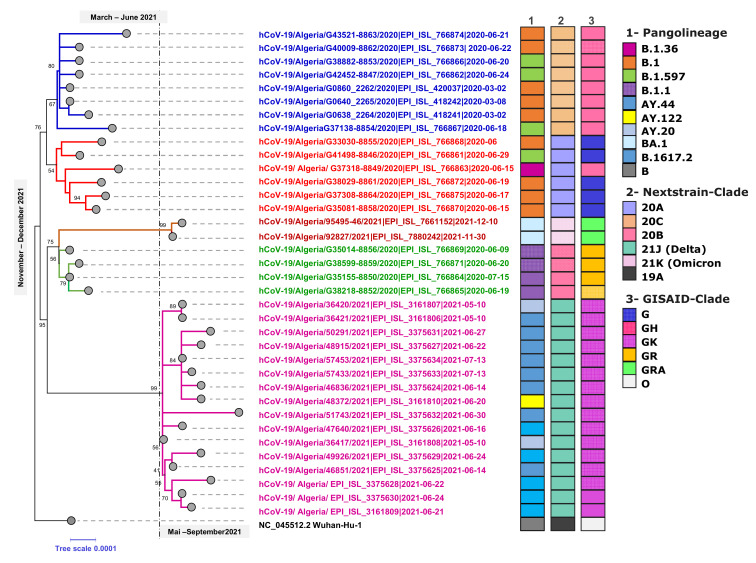
Rooted-Maximum likelihood phylogenetic tree of Algerian SARS-CoV-2 genome sequences (36 genomes). The genomes were classified into lineages using PANGOLIN clades, Nextclade and GISAID. The branch length on the phylogenetic tree represents the calendar time of sample collection: 18/36 samples were collected between March and June 2020, the 17/36 samples (21J—Delta) were collected late in May and July (2021) while the 2/36 samples were collected in November 2021 (21K-Omicron). The tree is rooted to the Wuhan reference genome (Wuhan/Hu-1/2019).

**Figure 3 microorganisms-10-00467-f003:**
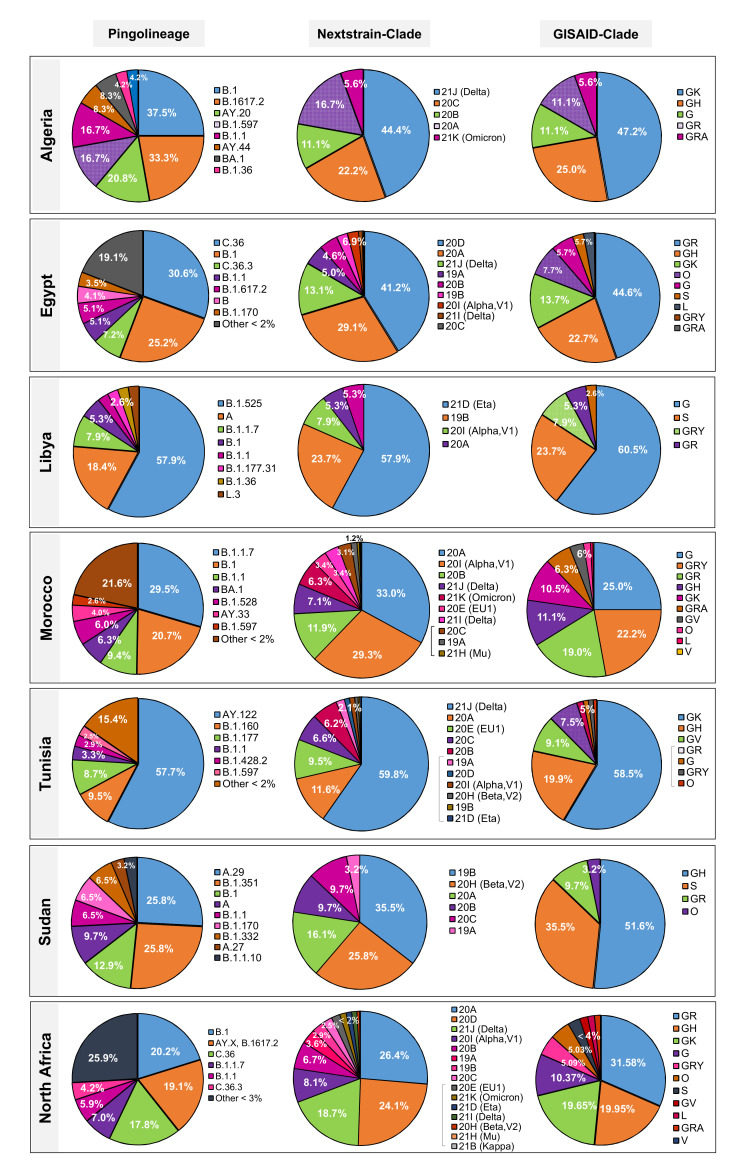
Distribution of various SARS-CoV-2 clades and lineages from Algeria and North African countries.

**Figure 4 microorganisms-10-00467-f004:**
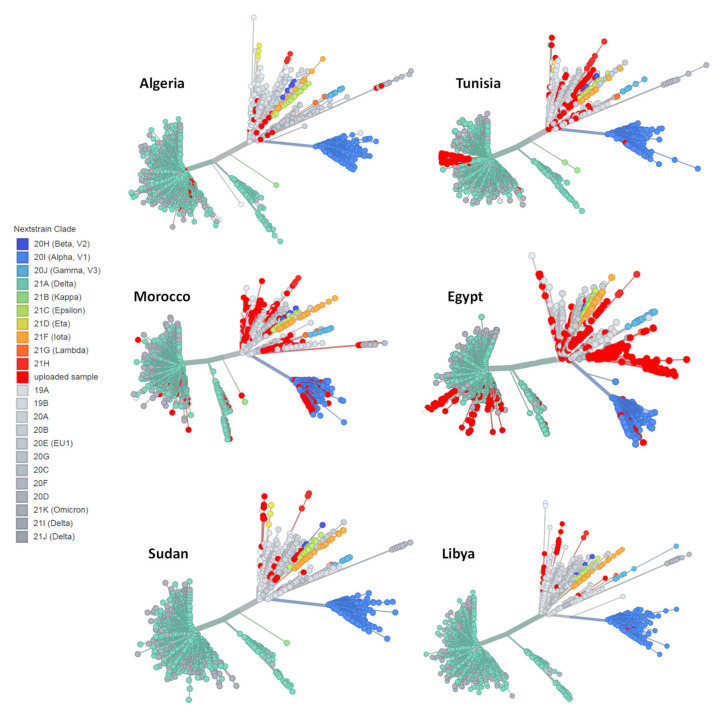
Unrooted phylogenetic relationship of analyzed SARS-CoV-2 sequences from North African countries to the global SARS-CoV-2 genomes. Clades distribution according to the Nextclade online tool (https://clades.nextstrain.org/) (accessed on 30 December 2021) and the phylogenetic tree was generated using the UShER Interface.

**Figure 5 microorganisms-10-00467-f005:**
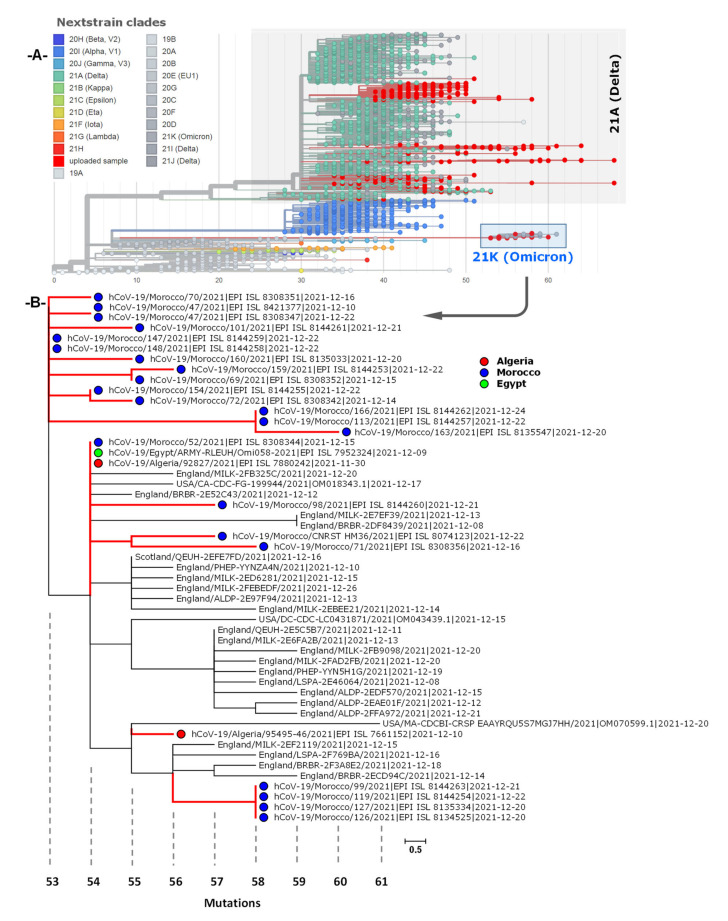
(**A**) Phylogenetic analysis generated by UShER interface of 350 SARS-CoV-2 sequences from North African countries. Subtrees including Omicron and Delta sequences plus 2031 random nearest sequences from the GISAID, GenBank, COG-UK, and CNCB databases (updated 30 December 2021). (**B**) Sequence comparison showing concordance, nucleic acid, and amino acid changes in the twenty-five Omicron sequences with respect to the SARS-CoV-2 Reference Sequence.

**Figure 6 microorganisms-10-00467-f006:**
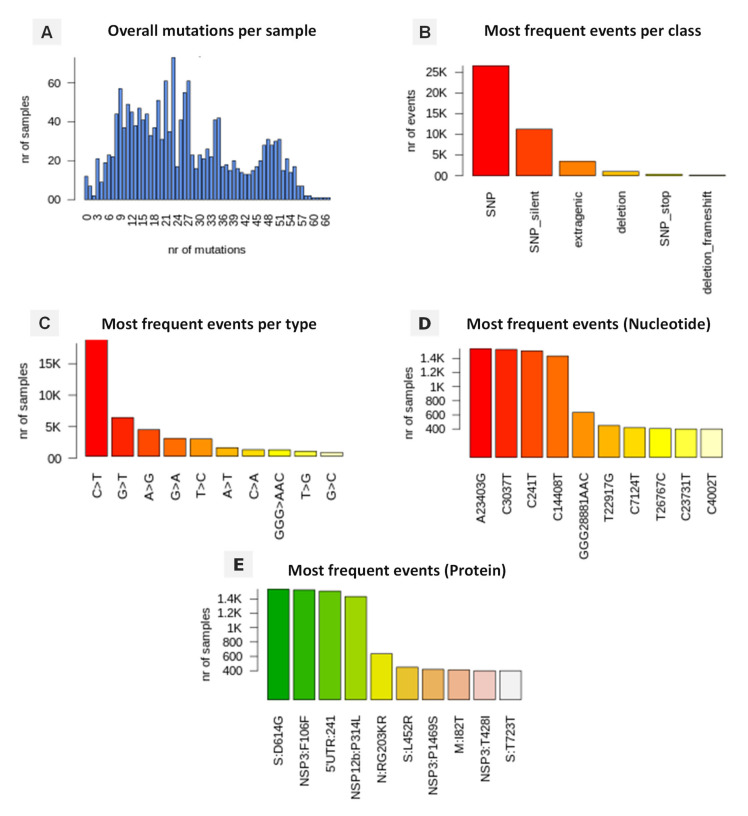
Distribution of numbers of mutational events of analyzed SARS-CoV-2 genomes in North African countries. (**A**) Distributions of number of mutations for each sample, (**B**) Distribution of SARS-CoV-2 mutation classes “SNP,” “deletion,” and “insertion”, (**C**) Listed nucleotide changes represent those found in the viral RNA, (**D**) distribution of SARS-CoV-2 most frequent specific events, annotated as nucleotide coordinates over the reference genome, (**E**) distribution of SARS-CoV-2 most frequent specific events, annotated protein changes using the format protein-mutation.

**Figure 7 microorganisms-10-00467-f007:**
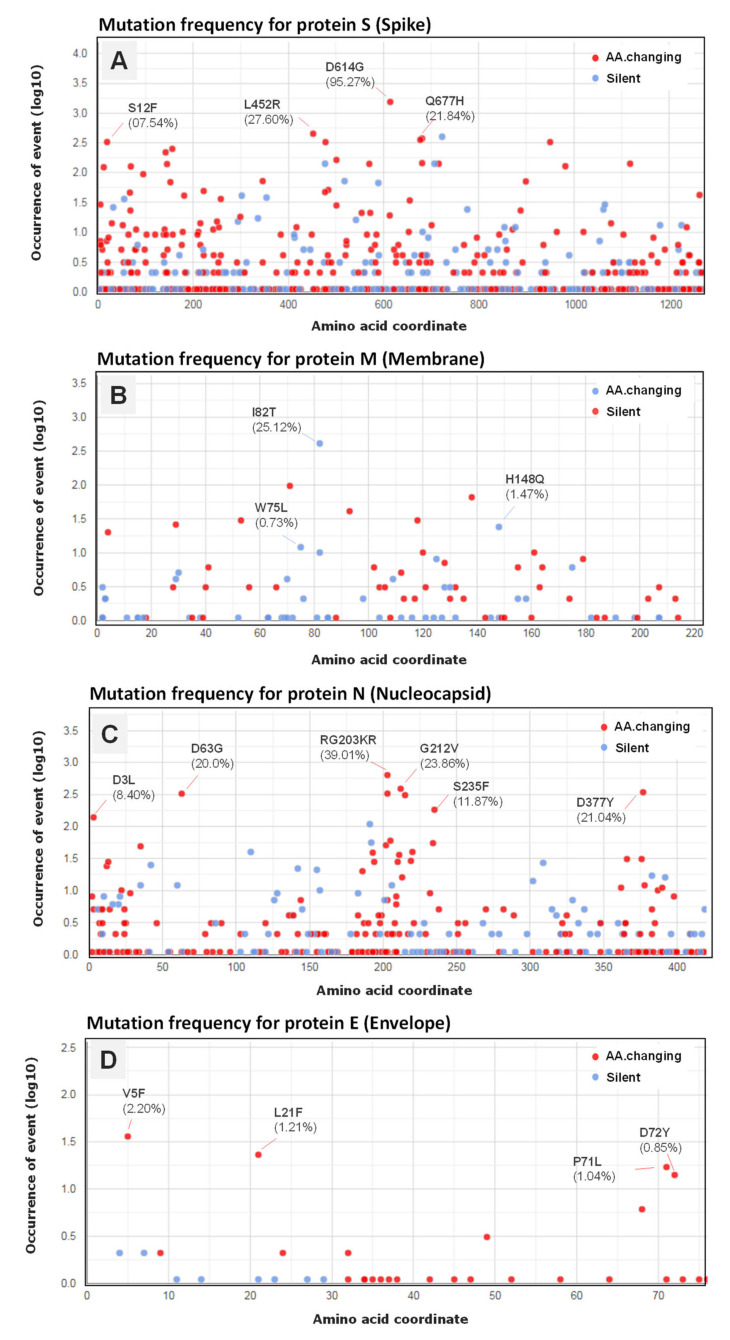
Occurrence of mutations in the SARS-CoV-2 structural proteins S (Spike) (**A**), M (Membrane) (**B**), N (Nucleocapsid protein (**C**), and E (Envelope (**D**). The frequency (in percentage) of the top amino acid-changing mutations is indicated.

**Table 1 microorganisms-10-00467-t001:** COVID-19 cases distribution and total analyzed SARS-CoV-2 genomes from North African countries ^1^.

Country	Total Cases	Sequenced Genomes	Analyzed Genomes
Algeria	214,592	85	36
Egypt	375,330	1418	971
Libya	381,749	56	38
Morocco	952,814	609	352
Sudan	45,112	116	31
Tunisia	721,031	315	241
Western Sahara	10	ND	ND
**Total**	**2,690,638**	**2599**	**1669**

^1^ 20 December 2021; ND No data available.

**Table 2 microorganisms-10-00467-t002:** The frequent mutation events observed in Omicron SARS-CoV-2 genomes.

Genomic Coordinate	Effect	N Samples	Class	Genomic Region
A23403G	S:D614G	25	SNP	Spike protein
G23948T	S:D796Y	25		
C23202A	S:T547K	25		
C24130A	S:N856K	25		
T23599G	S:N679K	25		
T24469A	S:N969K	25		
C23604A	S:P681H	25		
C24503T	S:L981F	25		
A24424T	S:Q954H	24		
C23854A	S:N764K	24		
C23525T	S:H655Y	24		
G22813T	S:K417N	24		
G22578A	S:G339D	24		
T22195G	S:N211K	23		
A23040G	S:Q493R	22		
TC22673CT	S:S371L	22		
T22679C	S:S373P	22		
C22686T	S:S375F	22		
G22992A	S:S477N	22		
C22995A	S:T478K	22		
A23013C	S:E484A	22		
G22898A	S:G446S	21		
C10029T	NSP4:T492I	25	SNP SNP	Transmembrane protein
A11537G	NSP6:I189V	25		
11286TGTCTGGTT	NSP6:L105	24	Deletion	
G8393A	NSP3:A1892T	25	SNP	Predicted phosphoesterase
A2832G	NSP3:K38R	24	SNP	
6513GTT	NSP3:S1265	24	Deletion	
C28311T	N:P13L	24	SNP	Nucleocapsid protein
GGG2881AAC	N:RG203KR	24		
G26709A	M:A63T	24	SNP	Membrane
A26530G	M:D3G	22		
C26577G	M:Q19E	24		
C26270T	E:T9I	25	SNP	Envelope
A18163G	NSP14:I42V	22	SNP	3′-to-5′ exonuclease
C10449A	NSP5:P132H	24	SNP	3C-like proteinase
C14408T	NSP12b:P314L	25	SNP	RNA-dependent RNA polymerase
A28271T	3′UTR:28271	25	Extragenic	3′ Untranslated Region
C241T	5′UTR:241	24	Extragenic	5′ Untranslated Region

SNP. Single-nucleotide polymorphism.

## Data Availability

The sequences analyzed in this study were downloaded from the GISAID database (https://www.gisaid.org) (accessed on 16 December 2021). The sequence metadata and other related documents are available as Appendix A.

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
