# Peer review of "Genomic Diversity of SARS-CoV-2 in Algeria and North African Countries: What We Know So Far and What We Expect?"

_microorganisms, 2022, doi:10.3390/microorganisms10020467_

Round 1
Reviewer 1 Report
Menasria et al. reported a first comprehensive genomic analysis of SARS-CoV-2 variants circulating in North African countries. The results should be useful for the current SARS-CoV-2 epidemic situation. The authors should answer the following questions before I give a decision. 1.The abstract is too long and should be simplified to highlight the novelty and emphasize the main findings of this manuscript. 2.Abstract GR, GH, GK, Line 89 VOC, VOI, VOHC etc. should be given their full names when their first appearance. 3.More explanation should be added with respect to “what we expect” in the title. 4.Figure 2, why two methods are used to do the clade analysis. It would be helpful to explain why there is slight difference by using two methods. Please add the meanings of symbols 20A etc. 5.Figures 4-6, I think they are not so clear enough for publication. 6.Discussion part: Current status on the rapid biosensing technologies for SARS-CoV-2, such as Journal of Hazardous Materials 425 (2022) 127923, Biosensors and Bioelectronics 190 (2021) 113418, are suggested to be added for their potential impact on the SARS-CoV-2 control.Author Response
Reviewer 1
Menasria et al. reported a first comprehensive genomic analysis of SARS-CoV-2 variants circulating in North African countries. The results should be useful for the current SARS-CoV-2 epidemic situation. The authors should answer the following questions before I give a decision.
- The abstract is too long and should be simplified to highlight the novelty and emphasize the main findings of this manuscript.
----Response.
We greatly appreciate the reviewer’s efforts and suggestions made. We thank the reviewer for the time dedicated to providing valuable feedback to our manuscript. As per suggestion, the abstract has been modified in a more straightforward way and full revision of the manuscript has been made to highlight its novelty and emphasize the main findings.
- Abstract GR, GH, GK, Line 89 VOC, VOI, VOHC etc. should be given their full names when their first appearance.
----Response.
Fixed as per suggestion, and the description below can be found in the revised manuscript - Page 1:L17-21
“Eleven GISAID clades with GR (variant of the spike protein S-D614G and nucleocapsid protein N-G204R), GH (variant of the ORF3a coding protein ORF3a-Q57H) and GK (variant S-T478K) being the most common with 25.9%, 19.9%, and 19.6% respectively followed by their parent clade G (variant S-D614G) (10.3%). Lower prevalence was noted for GRY (variant S-N501Y) (5.1%), S (variant ORF8-L84S) (3.1%) and GV (variant of the ORF3a coding protein NS3-G251V) (2.0%)”.
- More explanation should be added with respect to “what we expect” in the title.
----Response.
We thank the reviewer for the constructive remark. As per suggestion details with respect to “what we expect” in the title were added. The descriptions below can be found in the main manuscript and we have supplemented the reference accordingly.
- Page 14 L353-359 “Due to the naturally expanding genetic diversity of hCoV-19 viruses, extensive molecular surveillance and efforts to understanding the patterns of the global spread of the pandemic have been introduced including the three main nomenclatures PANGO lineages (PANGO, Phylogenetic Assignment of Named Outbreak LINeages) [19], Nextstrain clades [20], and GISAID classification. While PANGO lineages provide more detailed outbreak cluster information, the other two nomenclatures offer broad geographical and temporal clade trends.
- Page 16 L448-469 “The broad SARS-CoV-2 lineage diversity circulating in North African countries could intensify the impact of the pandemic in the region, affecting the efficacy of vaccines and display reduced antibodies neutralization even the reliability of diagnosis schemes including the current primary method of detecting SRAS-CoV-2 (Reverse transcription-quantitative polymerase chain reaction (RT-qPCR)) [48, 49]. However and within a very short period of time, research applied to COVID-19 diagnosis has advanced with ever-increasing knowledge and inventions, in adapting available virus detection technologies and exploiting the power of interdisciplinary research to design novel diagnostic tools or improve the detection efficiency [51, 52]. Given the epidemiological behaviors, current evidence supports that VOCs including Delta and the newly emerged Omicron variant have rapidly escalated, becoming predominant in the globe replacing previously circulating variants (https://nextstrain.org/ncov/gisaid/global) (accessed on 12 February 2022), adding up to a complex epidemiological scenario.
Compared with other variants and early identified SARS-CoV-2 strain, the high frequency of mutations in the spike sequence of the Omicron variant raises concern about a potential immune escape and its impact remains to be determined [53]. However, a complete experimental evaluation of Omicron might take weeks or even months. Large-scale case-control studies are essential to investigate clinical severity and the current situation must lead national governments to place a higher priority on timely collection and analysis. In fact, COVID-19 severity vary enormously depending on the country, the prevalence of vaccination, the population’s characteristics and medical management guidelines [54]”.
- Figure 2, why two methods are used to do the clade analysis. It would be helpful to explain why there is slight difference by using two methods. Please add the meanings of symbols 20A etc.
----Response.
We sincerely thank reviewer for the comments and recommendations. In this report and for the local Algerian virus comparison, only thirty-five sequences met the selection criterions and the SARS-CoV-2 genomes were classified into GISAID clades, PANGOLIN lineages and Nextclade. Due to the naturally expanding genetic diversity of hCoV-19 viruses, GISAID introduced a nomenclature system for major clades based on marker mutations within high-level phylogenetic groupings from the early split of S and L, to the further L into V and G, and later of G into GH, GR and GV, and more recently GR into GRY. Furthermore, GISAID clades are augmented with more detailed lineages assigned by the Phylogenetic Assignment of Named Global Outbreak LINeages (Pango lineage) tool, aiding in the understanding of patterns and determinants of the global spread of the pandemic. Later on, new major clades were proposed “Nextstrain” when reaches a frequency of 20% globally taken a count “sampling of sequences in time and space”. While PANGO lineages provide more detailed outbreak cluster information, the other two nomenclatures offer broad geographical and temporal clade trends. A clade name consists of the year it emerged (2019) and the available letter in the alphabet (A) for example so the clades were named 19A, 20A, 20B…etc. A new clade should be at least 2 mutations away from its parent major clade.
In addition, only variants of concern (VOC) and variants of interest (VOI) were named by WHO including: B.1.1.7/GRY/21I as (Alpha), B.1.351/GH/20H (Beta), P.1/GR/20J (Gamma), B.1.617/G/21A (Delta) and B.1.1.529/GRY/21K as (Omicron). VOIs include variants B.1.427/9 (Epsilon), B.1.525 (Eta), B.1.526 (Iota), B.1.617.1 (Kappa), C.37 (Lambda) and Mu (B.1.621).
- Figures 4-6, I think they are not so clear enough for publication.
----Response. The quality of the manuscript according to the journal format was improved. For the potential readers, the clarity of the figures was fixed and all suggested changes have been incorporated.
- Discussion part: Current status on the rapid biosensing technologies for SARS-CoV-2, such as Journal of Hazardous Materials 425 (2022) 127923, Biosensors and Bioelectronics 190 (2021) 113418, are suggested to be added for their potential impact on the SARS-CoV-2 control.
----Response. As per suggested we have formatted the references and adapted suggested one in the revised manuscript. The description below can be found in the main manuscript and we have supplemented the reference accordingly.
- Page 16 L448- 460 “The broad SARS-CoV-2 lineage diversity circulating in North African countries could intensify the impact of the pandemic in the region, affecting the efficacy of vaccines and display reduced antibodies neutralization even the reliability of diagnosis schemes including the current primary method of detecting SRAS-CoV-2 (Reverse transcription-quantitative polymerase chain reaction (RT-qPCR)) [49, 50]. However and within a very short period of time, research applied to COVID-19 diagnosis has advanced with ever-increasing knowledge and inventions, in adapting available virus detection technologies and exploiting the power of interdisciplinary research to design novel diagnostic tools or improve the detection efficiency [51, 52]”
- Yang, Y., Liu, J., Zhou, X. A CRISPR-based and post-amplification coupled SARS-CoV-2 detection with a portable evanescent wave biosensor. Biosensors and Bioelectronics. 2021. 190, 113418. doi:10.1016/j.bios.2021.113418
- Zhu, Q., & Zhou, X. A colorimetric sandwich-type bioassay for SARS-CoV-2 using a hACE2-based affinity peptide pair. Journal of Hazardous Materials. 2022. 425, 127923. doi:10.1016/j.jhazmat.2021.127923
Reviewer 2 Report
This is a well conceived and written work. The main topic has a significant importance in Covid-19 research. The practical issue of this work appears to be quite relevant at present and in the future monitoring of SARS-CoV-2 genomic variability. The theoretical background is sufficiently contextualized in the introduction. Description of methods is exhaustive. The quantitative findings are adequately summarized in “Results” section text and the discussion is coherent with them. I have no significant further suggestions.
Author Response
Reviewer 2
This is a well-conceived and written work. The main topic has a significant importance in Covid-19 research. The practical issue of this work appears to be quite relevant at present and in the future monitoring of SARS-CoV-2 genomic variability. The theoretical background is sufficiently contextualized in the introduction. Description of methods is exhaustive. The quantitative findings are adequately summarized in “Results” section text and the discussion is coherent with them. I have no significant further suggestions.
----Response. We greatly appreciate the reviewer’s efforts to carefully review the manuscript. We really appreciate the time dedicated to providing valuable and positive feedback to our manuscript.
The quality of the manuscript according to the journal format has been improved. The clarity of the figure legends has been fixed and improved, and English was double-checked, improved and grammatically edited. All suggested changes to wording, typos and punctuation have been incorporated.
Reviewer 3 Report
Congratulations to the authors for the very interesting article.
The present article entitled ”Genomic diversity of SARS-CoV-2 in Algeria and North African countries: What we know so far and what we expect?”
I woul suggest the authors to take some informations also form this articles because they mention interestings things about the pandemic:
DOI 10.2147/RMHP.S284557, 10.3390/microorganisms9040793, 10.3390/microorganisms8111704
In the discussion chapter please compare your results with the results from other similar studies.
Please provide also a conlusion for the study.
Author Response
Reviewer 3
Congratulations to the authors for the very interesting article.
- The present article entitled”Genomic diversity of SARS-CoV-2 in Algeria and North African countries: What we know so far and what we expect?”
I would suggest the authors to take some information also form these articles because they mention interesting things about the pandemic:
DOI 10.2147/RMHP.S284557, 10.3390/microorganisms9040793,
----Response. We greatly appreciate the reviewer’s efforts and time dedicated to providing valuable and positive feedback to our manuscript. As per suggested, we have formatted the references and adapted the suggested one in the revised manuscript. The description below can be found in the main manuscript and we have supplemented the reference accordingly.
- Page 15, L426-431 “The spike glycoprotein region mediates infection of target cells through binding to its cognate receptor angiotensin converting enzyme 2 (ACE2) and initiating viral-host fusion and replication [41]. This region is reported to be the most essential for viral attachment and entry into the host cells [42, 43]. Therefore, ACE2 expression in different tissues and interaction with SARS-CoV-2 are critical for the infection progression to severe coronavirus disease 2019 (COVID-19) [43].
43. Yalcin, H. C., Sukumaran, V., Al-Ruweidi, M. K. A., & Shurbaji, S. Do changes in ace-2 expression affect sars-cov-2 virulence and related complications: A closer look into membrane-bound and soluble forms. International Journal of Molecular Sciences, 2021, 22(13), 6703. doi.org/10.3390/ijms22136703
- In the discussion chapter, please compare your results with the results from other similar studies.
Please provide also a conclusion for the study.
----Response.
We thank the reviewer for the valuable and insightful remarks. Findings from the present investigation were interpreted (section Discussion) and fixed as per suggestion in a more straightforward way in relation to what was already known about the research problem, and compare the results with other studies to support our claim. In addition, conclusions are under the discussion section in separated one.